

# Transcriptome and metabolome profiling reveal the inhibitory effects of food preservatives on pathogenic fungi

Zhenxia Shi[1], Ni Zhan[1], Ming Ma[2], Zhen Wang[1], Xunyou Yan[1], Rumeng Li[1] and Xuejuan Liu[1]

[1] College of Life Science, Langfang Normal University, Langfang, Hebei, China
[2] Langfang Agricultural Information Center, Langfang, Heibei, China

## ABSTRACT

Sec-butylamine, potassium sorbate, and citric acid were selected as preservatives to investigate their inhibitory effects on common plant pathogens—*Aspergillus flavus*, *Alternaria alternata*, and *Talaromyces funiculosus*—as well as the inhibitory mechanism of sec-butylamine against *A. flavus*. The results showed that all three preservatives significantly inhibited the growth of the tested fungi. Under the experimental conditions, 0.6% sec-butylamine, 1.2% citric acid, and 0.2% potassium sorbate completely inhibited the growth of *A. flavus*. Similarly, 0.5% sec-butylamine, 1.0% citric acid, and 0.6% potassium sorbate completely inhibited *A. alternata*, while 1.0% sec-butylamine, 1.2% citric acid, and 0.8% potassium sorbate completely inhibited *T. funiculosus*. All three preservatives exhibited strong inhibitory activity against mycelial growth, with inhibition increasing alongside concentration under *ex vivo* conditions. To explore the inhibitory mechanism of sec-butylamine on *A. flavus*, transcriptomic and metabolomic analyses were conducted on *A. flavus* mycelia before and after treatment. The results revealed that key genes such as AFLA_053390, AFLA_121370, AFLA_024930, and AFLA_041970 were significantly downregulated following sec-butylamine exposure. Additionally, AFLA_002830 and AFLA_030450 also showed reduced expression levels. Metabolomic analysis identified several metabolites associated with sec-butylamine treatment. Compounds such as (3R)-4,4-dimethyl-2-oxotetrahydro-3-furanyl β-D-glucopyranoside (Com_5857_neg), trehalose (Com_3182_neg), D-glucosamine 6-phosphate (Com_4401_neg), and sucrose (Com_494_neg) were elevated, while D-gluconic acid (Com_9540_neg), D-glucose 6-phosphate (Com_723_neg), verbascose (Com_11501_neg), and D-(-)-fructose (Com_285_neg) were reduced after treatment. This study provides a reference for the practical application of food preservatives and lays a foundation for further research into their antifungal mechanisms.

## INTRODUCTION

The issue of food safety had become one of the global concern for public health, and food-borne diseases resulting from food-spoiling pathogenic microorganisms constitute key factors in food safety problems (*You et al., 2022*). Whilst chemical food preservatives were currently the most widely employed means of food preservation worldwide, chemical

Corresponding author
Zhen Wang, 1232134@lfnu.edu.cn

preservatives often caused disadvantages such as pesticide residue, environmental pollution and heightened resistance of pathogenic fungi. Common pathogenic fungi in fruits and vegetables included *Alternaria alternata* (*Choi et al., 2002*) and *Aspergillus flavus* (*Abhishek et al., 2021*), which were significant and prevalent toxin-generating fungi infecting food crops, whilst contamination of crops with their toxic metabolites was rather commonplace, and human health was profoundly impacted by them. Whilst national health authorities rigorously supervise food crops containing their toxins, it remained challenging to regulate food crops with low toxin content. Hence, there existed an urgent requisite for the food industry to develop preservation methods that were safe, efficient to implement, inexpensive and boast a wide range of applications. *Talaromyces funiculosus* belonged to the genus Talaromyces, which were widely dispersed in air and soil, some strains represented beneficial species for industrial production, and were extensively employed as enzyme and pigment generating fungi, and some of which were detrimental strains that caused food spoilage and induce conditional pathogens (*Hai et al., 2021*). However, research into the inhibition of *T. funiculosus* was not extensive, and the present experiment would utilize a variety of preservatives to conduct preliminary inhibition tests on *T. funiculosus* in order to enrich the experimental data for inhibition studies on them.

Common food preservation methods included wet sand storage, wax seal storage, gas storage, and chemical reagent preservation (*Pino-Hernández et al., 2021*). However, due to difficulties in operation, applicable scale, transportation and cost, the use of chemical preservatives for food eventually emerged as an economical, long-lasting and practical approach (*Sharif et al., 2017*). Many countries have discovered that chemical preservatives provided advantages over traditional techniques and were extensively using, researching and developing chemical preservatives in the food industry (*Yuan et al., 2023*). Common chemical food preservatives included sodium benzoate, benzoic acid, potassium sorbate (*Lin et al., 2018*), calcium propionate, citric acid (*Yang et al., 2019*) and others. Among those, citric acid and potassium sorbate as chemical preservatives were food additives approved by national standards. They were economical, practical and less harmful to the human body (*Latif et al., 2022*; *Soccol et al., 2006*; *Faheem et al., 2022*). Sec-butylamine was a fungicide with significant inhibitory effects on various molds. After Dr. Eckert discovered its fungicidal properties in 1962, it became widely used for preserving citrus, grapes, apples and peppers (*Felicia et al., 2022*; *Wang et al., 2022*). In China, sec-butylamine was approved in 1987 as a post-harvest preservative for fruits and vegetables (*Eckert, 1963*). Since sec-butylamine was first recommended for use in 1975 and was now widely applied in many countries for storing fruits and vegetables, the exploration and development of sec-butylamine preservatives in the food industry has continued. However, there has been no report on the inhibitory effect and mechanism of sec-butylamine on *A. flavus*.

The main purpose of this study was to investigate the inhibitory effects of different concentrations of sec-butylamine, potassium sorbate and citric acid on common pathogenic fungi in fruits and vegetables, namely *A. flavus*, *A. alternata*, *T. funiculosus* and the transcriptome and metabolome analysis were used to explore the inhibitory mechanism of sec-butylamine against *A. flavus*. The aim was to provide data as a reference

for their application in actual preservation practices and establish a foundation of data for subsequent studies.

## MATERIALS & METHODS

### Materials

*Aspergillus flavus* (1-b2-1.3), *Alternaria alternata* (2-c2-4.2), and *Talaromyces funiculosus* (4-a3-2.1) were obtained from the Biotechnology Research Laboratory of Langfang Normal University.

### Spore suspension preparation and plate culture

The mold preserved on the PDA slant medium was inoculated on the PDA plate (*Lin et al., 2018*) and incubated at 30 °C during seven days. Sterile water was added with a sterile triangular glass rod to gently scrape the spores and mycelium on the medium to make part of the mycelium suspended in the water layer. The water layer was then transferred to a 50 ml triangular conical flask and 15∼25 glass beads of about six mm in diameter were added. The flask was shaken at 30 °C for about 3 mins, and then the mold containing spores and mycelium suspension was filtered off the mycelium with three layers of sterilizing gauze. The resulting spore suspension was counted by hemocytic technique plate, and the spore suspension concentration was controlled in the range of $10^5$∼$10^6$ CFU/mL (*Dantigny & Nanguy, 2009*). 0.2 ml of the above spore suspension was evenly coated onto PDA medium and incubated in a constant temperature incubator at 28 °C during 3 days. Fungus with a diameter of six mm were punched out on the Petri dishes with a sterile punch and set aside.

### The effect of preservatives on the growth of the test strain

Two ml of different concentrations of food preservative solution were mixed with 100 ml of PDA medium and poured into sterile Petri dishes, the medium was retained for naturally cooling and put in the incubator, 28 °C during 24 h to ensure sterility of plate. The uncontaminated medium was taken and placed a piece of beaten fungal mass in the center of the Petri dish. Three parallel experiments were conducted for each treatment, with equal amount of sterile water in the medium as the control group. The culture was incubated in a constant temperature incubator at 28 °C during 36 h. The colony diameter was measured by the crossover method and the inhibition rate was calculated (*Schutt & Netzly, 1991*; *Lederer, Lorenz & Seemüller, 1992*).

$$\text{Inhabition rate(\%)} = (1 - \text{net growth of treatment/net growth of control}) \times 100$$

where, net growth = colony diameter (mm) − disc diameter (6 mm).

### Regression equation of the inhibition of the preservative to the test strains

The mycelial growth rate method (*Raimbault & Alazard, 1980*; *Smith & Walker, 1981*) was used to determine the magnitude of inhibition of the three conservatives against each of the three pathogenic fungi. According to the biostatistical chance value conversion table, the inhibition rates were converted into chance values and the logarithm of the concentrations

was taken. The vertical coordinate was the chance value and the horizontal coordinate was the logarithm of the concentration, and the regression equation and value of the virulence of the three preservatives against each of the three pathogenic fungi were derived using an Excel sheet (*McClelland, Bernhardt & Casadevall, 2006*).

## Scanning electron microscope observation

After collected fresh mycelium and fixed it in 2.5% glutaraldehyde at 4 °C during 12 h, the samples were dehydrated in 70% ethanol during 15 mins, dehydrated in 90% and anhydrous ethanol during 20 mins, respectively. The samples were then transferred to 25%, 50%, 75%, and 100% concentrations of acetone during 10 mins each (*Lima, Colombo & de Almeida Junior, 2019*). After that, the samples were successively transferred to an elution buffer containing isoamyl acetate and acetone during 10 mins. Finally, the samples were dried and fixed, and the sample was observed by scanning electron microscope (SEM) after the ion sputtering sprayer sprayed the samples with gold.

## Transcriptome analysis

Three biological replicates of sec-butylamine treatment of *A. flavus* and control under the same growth conditions were used for transcriptome sequencing. Z1, Z2, and Z3 represented treatment samples, and ZC1, ZC2, and ZC3 represented control samples. Total RNA was extracted using the Ezup Column Fungi Genomic DNA Purification Kit (Sangon Biotech, Shanghai, China). The RNA concentration was measured using Nanodrop, and RNA purity and integrity were assessed by 1% agarose gel electrophoresis. 28S/18S and RNA integrity number (RIN) were analyzed using the RNA Nano 6000 Assay Kit of the Bioanalyzer 2100 system (Agilent Technologies, Santa Clara, CA, USA). Agilent 2100 and qPCR were used for quality control. The libraries were subjected to transcriptome sequencing using the Illumina Novaseq platform.

Raw data (raw reads) of fastq format were firstly processed through fastp software, the raw data were filtered for reads with junctions and all-A bases and for more than 10% N- or low-quality reads. Hisat2 v2.0.5 was used to compare the filtered reads with the NRRL 3357 (NCBI accession number: EQ963471) reference genome for analysis. Feature Counts v1.5.0-p3 was used to count the reads numbers mapped to each gene. The Fragments Per Kilobase of exon model per Million mapped fragments (FPKM) of each gene was calculated based on the length of the gene and reads count mapped to this gene (*Trapnell et al., 2012*), and differential analysis of mRNAs across samples were performed using DESeq2 R package (1.20.0) (*Robinson, Mccarthy & Smyth, 2010*), with a screening condition of FDR (false discovery rate) < 0.05 and |log2FC (fold change)| > 1.

The clusterProfiler R package was used to analyze GO (Gene Ontology) enrichment of differentially expressed genes. We used clusterProfiler R package to test the statistical enrichment of differential expression genes in the Kyoto Encyclopedia of Genes and Genomes (KEGG) pathways.

## Reverse transcription of RNA and q RT-PCR

After extraction of RNA using the SUPERspin Plant RNA Rapid Extraction Kit (Vazyme, Nanjing, China), cDNA was synthesized using the HiScript III All-in-One RT SuperMix

Perfect for qPCR (Vazyme, Nanjing, China) reverse transcription kit. Primers for candidate genes were added to cDNA and Taq pro Universal SYBR qPCR Master Mix Kit for qRT-PCR analysis, which was performed in a fluorescent quantitative PCR instrument (Bio-Rad, Hercules, CA, USA) according to the manufacturer's instructions. For each sample, we employed three biological replicates using AFLA_nor as the reference gene. Conditions for PCR amplification were as follows: each cycle consisted of 90 s of pre-denaturation at 95 °C, 5 s of denaturation at 95 °C, 15 s of annealing at 60 °C, and 20 s of extension at 72 °C. After every cycle, which totaled 35 cycles and three replicates of each reaction, we obtained fluorescence signals. For the primer design, Primer Express 2.0 was used (Table S1). The $2^{-\Delta\Delta C}$ approach was employed for data analysis (*Livak & Schmittgen, 2001*).

## Metabolome analysis

The samples (one mL) were freeze-dried and resuspended with prechilled 80% methanol by well vortex. D1, D2, D3, D4, D5, and D6 represented treatment samples, and DC1, DC2, DC3, DC4, DC5, and DC6 represented control samples. Then the samples were incubated on ice during 5 mins and centrifuged at 15,000 g, 4 °C during 15 mins. Some of supernatant was diluted to final concentration containing 53% methanol by LC-MS grade water. The samples were subsequently transferred to a fresh Eppendorf tube and then were centrifuged at 15,000 g, 4 °C during 15 mins. Finally, the supernatant was injected into the LC-MS/MS system analysis. UHPLC-MS/MS analyses were performed using a Vanquish UHPLC system (Thermo Fisher Scientific, Waltham, MA, USA) coupled with an Orbitrap Q Exactive TM HF mass spectrometer or Orbitrap Q Exactive TMHF-X mass spectrometer (Thermo Fisher Scientific, Waltham, MA, USA) in Novogene Co., Ltd. (Beijing, China). Samples were injected onto a Hypersil Goldcolumn (100 × 2.1 mm, 1.9 μm) using a 12-min linear gradient at a flow rate of 0.2 mL/min. Q Exactive TM HF mass spectrometer was operated in positive/negative polarity mode with spray voltage of 3.5 kV, capillary temperature of 320 °C, sheath gas flow rate of 35 psi and aux gas flow rate of 10 L/min, S-lens RF level of 60, Aux gas heater temperature of 350 °C. UHPLC-MS/MS generated the raw data files, and perform peak alignment, peak picking, and quantitation for each metabolite were processed by the Compound Discoverer 3.3 (CD3.3, Thermo Fisher Scientific, Waltham, MA, USA). The KEGG database, HMDB database and LIPIDMaps database were used to annotate these metabolites. Principal component analysis (PCA) and partial least squares discriminant analysis (PLS-DA) were performed at metaX (*Wen et al., 2017*). The univariate analysis (*t*-test) was used to calculate the statistical significance (*P*-value). The difference criterion of metabolites was VIP > 1 and *P*-value < 0.05 and fold change ≥ 2 or FC ≤ 0.5.

## RESULTS

### Effect of three preservatives on the inhibition effect of the test strains

The effects of sec-butylamine, citric acid, and potassium sorbate on the inhibition of *A. flavus*, *A. alternata*, and *T. funiculosus* were shown in Fig. 1. It could be seen that the three preservatives have a strong inhibitory effect on the three pathogenic fungi, and the inhibition effects became more obvious as the concentration increases. Under the conditions of this experiment, six mg/ml of sec-butylamine, 12 mg/ml of citric acid, and two mg/ml of potassium sorbate had a significant inhibitory effect on the growth of *A. flavus*. Four mg/ml of sec-butylamine, 10 mg/ml of citric acid, and six mg/ml of potassium sorbate had a significant inhibitory effect on the growth of *A. alternata*. While 10 mg/ml of sec-butylamine, 12 mg/ml of citric acid, and eight mg/ml of potassium sorbate significantly inhibited the growth of *T. funiculosus*.

Sec-butylamine was an alkaline preservative that creates an alkaline growth environment for the strain, which affected the growth of *A. flavus*, *A. alternata*, and *T. funiculosus*. The higher the concentration of sec-butylamine, the stronger the alkalinity, and the more significant the inhibition effect (Fig. 1). Under the treatment of citric acid, the strain was in an acidic growth environment, which inhibited the growth and reproduction of *A. flavus*, *A. alternata*, and *T. funiculosus*. The higher the concentration of citric acid, the stronger the acidity, and the more significant the inhibition effect (Fig. 1). Potassium sorbate could inhibit the activity of dehydrogenase in pathogenic fungi and destroy various important enzyme systems of *A. flavus*, *A. alternata*, and *T. funiculosus*, effectively inhibiting their growth and reproduction, thus achieving the purpose of fungus inhibition and antisepsis. It could be seen that the three preservatives have a significant inhibitory effect on the three pathogenic fungi, respectively, and within a certain concentration range, the inhibitory effects of single preservatives with different concentrations and different preservatives with the same concentration on the pathogenic fungi are significant. The inhibition of mycelial growth of pathogenic fungi was enhanced with the increase of concentration (Fig. S1).

### Inhibition equations of the three preservatives against the test strains

Table S2 displaied the inhibition equations and $EC_{50}$ (effective concentrations) values of sec-butylamine, citric acid, and potassium sorbate against *A. flavus*, *A. alternata*, and *T. funiculosus*, respectively. It could be seen that the three preservatives have good growth inhibition effects on the three pathogenic fungi within the range of test concentrations. Under isolated conditions, the growth inhibition rate increased with the increase of the preservative concentration. The smaller the $EC_{50}$ value of the preservative to the pathogenic fungi, the stronger the inhibition effect on the mycelial growth of the pathogenic fungi. Through calculation and comparison, it was found that the $EC_{50}$ value of potassium sorbate to *A. flavus* was the smallest at 0.4819 mg/ml, indicating the strongest inhibition effect, which was consistent with the experimental results of the strain growth effect. The $EC_{50}$ value of sec-butylamine against *A. alternata* was the smallest at 1.0709 mg/ml, indicating the strongest inhibitory effect, which was consistent with the experimental results of the strain growth effect. The $EC_{50}$ value of potassium sorbate against *T. funiculosus* was the

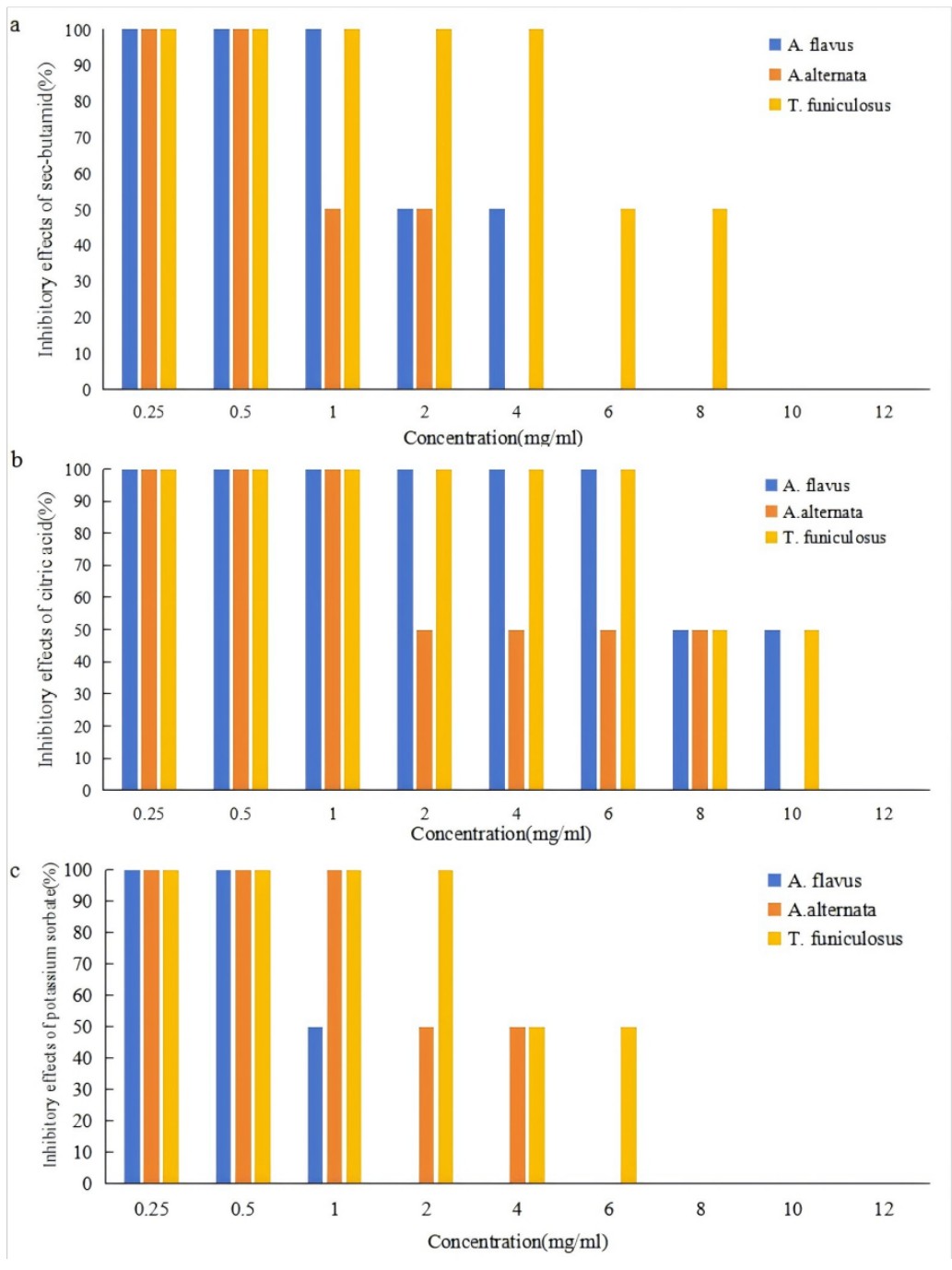

**Figure 1  Inhibitory effects of preservatives on *Aspergillus flavus*, *Alternaria alternata* and *Talaromyces funiculosus*.** (A) represented inhibitory effects of *Aspergillus flavus*, (B) represented inhibitory effects of *Alternaria alternata*, and (C) represented inhibitory effects of *Talaromyces funiculosus*.

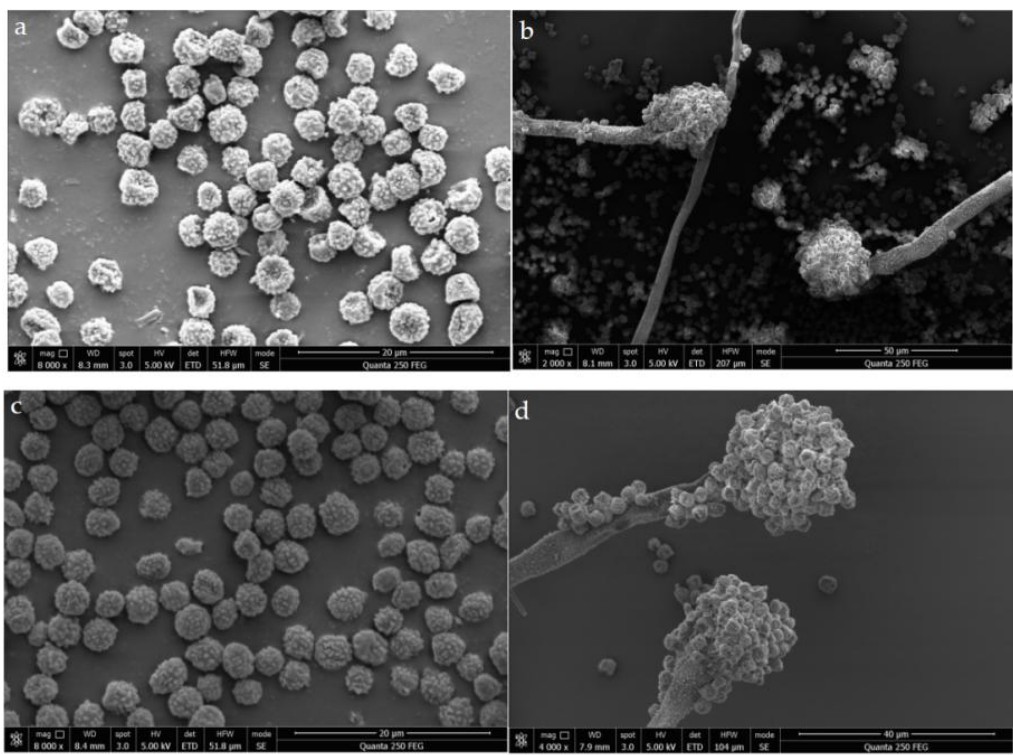

**Figure 2   SEM image of the microstructure of *Aspergillus flavus*.** (A, B) represented treatment samples, which were treated with sec-butylamine; (C, D) represented control samples.

smallest at 1.9216 mg/ml, indicating the strongest inhibitory effect, which was consistent with the experimental results of the strain growth effect (Table S2).

## Microstructure of *A. flavus*

In order to explore the change of the microscopic structure of *A. flavus* mycelium under the action of sec-butylamine, the microscopic structure of *A. flavus* cultured on solid medium during 7 days were observed by scanning electron microscopy. The microstructural changes of *A. mycelia* after treatment with sec-butylamine reflected the degree of inhibition of *A. flavus* mycelia growth by sec-butylamine. In the control group, the form of *A. flavus* mycelium was normal, and the mycelium was smooth and full. In the sec-butylamine treatment group, the morphology of *A. flavus* mycelia was changed, and spines appeared on the surface and some mycelia were dented (Fig. 2).

## Transcriptome analysis of *A. flavus*

The transcriptome and metabolome analysis of *A. flavus* mycelia before and aftersec-butylamine treatment were performed to further explore the inhibitory mechanism of sec-butylamine on *A. flavus* in this study.

Transcriptome sequencing identified 273,675,516 raw reads and 41.05G raw bases, 270,194,450 clean reads and 40.52G clean bases after filtering out inferior reads. The Q30 ranged from 94.64% to 95.28%, while the GC content varied from 51.18% to 53.05%

**Table 1 Differentially expressed genes related to sec-butylamine inhibits *Aspergillus flavus*.**

| Gene_id | Log2 fold change | *P* value | Gene_description |
|---|---|---|---|
| AFLA_053390 | 0.944216706 | 3.97E−05 | beta-1,3-endoglucanase |
| AFLA_121370 | 1.368574152 | 0.00288219 | 1,3-beta-glucanosyltransferase |
| AFLA_024930 | 0.563523481 | 0.02821282 | extracellular endoglucanase |
| AFLA_041970 | 0.60472489 | 0.029250377 | chitin binding protein |
| AFLA_002830 | 0.712771848 | 0.032973313 | alpha-trehalose-phosphate synthase subunit |
| AFLA_030450 | 0.707742767 | 0.040524036 | trehalose synthase |

for all samples, and these results demonstrate the high quality of the sequencing data obtained (Table S3). Furthermore, evaluating the sequencing data against the *A. flavus* genome revealed that the mapped reads ranged from 91.99% to 93.21%, and the Uniq mapped reads ranged from 91.43% to 92.79% (Table S4). PCA analysis of the samples showed that samples of the same treatments aggregated together while different treatments among different varieties could be differentiated, and there were differences between Z1, Z2, Z3 and ZC1, ZC2, ZC3, implying the reliability of our sequencing data (Fig. S2). We detected 967 DEGs (differentially expressed genes) in Z *vs.* ZC groups, including 638 with up-regulation and 329 with down-regulation (Fig. S3).

The identified DEGs were broadly categorized according to their biological process, and molecular function. Within biological processes, DEGs were mainly associated with cellular response to chemical stimulus, response to chemical, cellular response to toxic substance, and cellular oxidant detoxification. Within cellular components, DEGs were categorized in terms of ribosomal subunit. In molecular function, DEGs were generally associated with anion transmembrane transporter activity and organic acid transmembrane transporter activity (Fig. S4). KEGG enrichment analysis demonstrated that DEGs in Z *vs.* ZC were mostly related to metabolic pathway, biosynthesis of secondary metabolites and peroxisome (Fig. S5).

## Key genes involved in sec-butylamine inhibits *A. flavus*

To study the key genes of sec-butylamine inhibits *A. flavus*, the differentially expressed genes were analyzed. AFLA_053390 encoded beta-1,3-endoglucanase, after sec-butylamine treatment, the expression decreased. AFLA_121370 encoded 1,3-beta-glucanosyltransferase, and the expression was down-regulated after treatment in *A. flavus*. AFLA_024930 encoded extracellular endoglucanase, after treatment with sec-butylamine, the expression was down-regulated in *A. flavus*. AFLA_041970 encoded chitin binding protein, the expression was down-regulated after treatment. AFLA_002830 encoded alpha-trehalose-phosphate synthase subunit and AFLA_030450 encoded trehalose synthase, both of them were treated with sec-butylamine and their expression levels decreased in *A. flavus* (Table 1). Q RT-PCR was used to confirm key genes associated with sec-butylamine inhibits *A. flavus*, and the results were in line with the transcriptome result. They showed the same trend in different samples (Fig. 3).

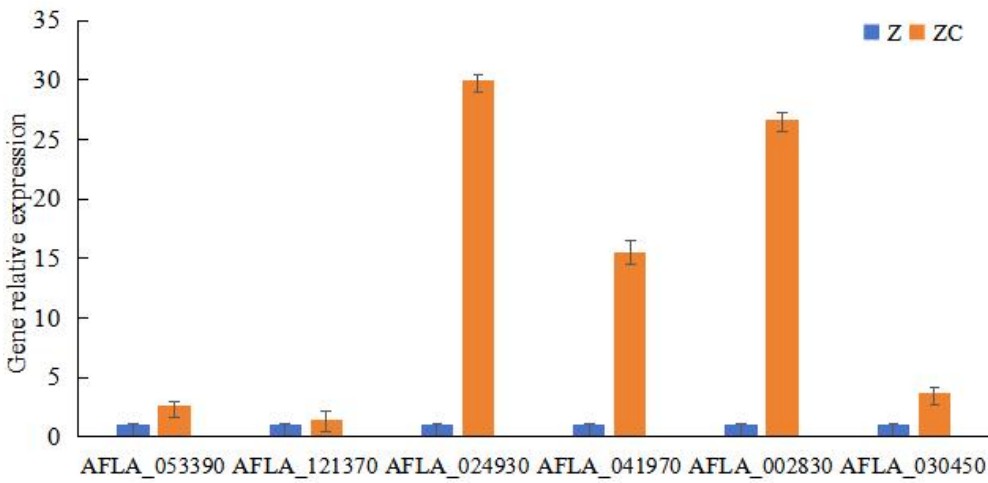

**Figure 3** **Results of qRT-PCR for genes related to melon wilt resistance.** Z represented treatment samples, and ZC represented control samples.

## Metabolome analysis of *A. flavus*

PCA analysis classified overall variation as PC1 and PC2, contributing 23.56/22.61%, respectively (Fig. 4). The results of metabolome analysis revealed that metabolites of *A. flavus* treated with sec-butylamine were significantly different from control samples (Fig. 5).

A total of 108 differential metabolites were detected in the D and DC samples based on the UPLC-MS/MS detection platform and the local metabolic database (Biomarker Technologies, Beijing, China) (Table S5). Furthermore, the KEGG pathway enrichment analysis showed that the significantly enriched pathways were metabolic pathway, biosynthesis of antibiotics, purine metabolism, starch and sucrose metabolism (Fig. 6).

## Key metabolites involved in sec-butylamine inhibits *A. flavus*

We further analyzed metabolites associated with sec-butylamine inhibits *A. flavus*. (3R)-4,4-Dimethyl-2-oxotetrahydro-3-furanyl beta-D-glucopyranoside (Com_5857_neg), Trehalose (Com_3182_neg), D-Glucosamine 6-phosphate (Com_4401_neg), as well as Sucrose (Com_494_neg) present at high levels after sec-butylamine treatment in *A. flavus*. In contrast, D-Gluconic acid (Com_9540_neg), D-Glucose 6-phosphate (Com_723_neg), Verbascose (Com_11501_neg), and D-(-)-Fructose (Com_285_neg) present at low levels after sec-butylamine treatment in *A. flavus* (Fig. 7).

## DISCUSSION

According to studies, there were almost 100 foodborne pathogenic microorganisms found in fresh vegetables (*Nassarawa, Luo & Lu, 2022*). Moreover, some studies suggested that some studies suggested that the global underreporting of foodborne diseases could be as high as 90% (*Zhang et al., 2022*). These findings suggested that many countries worldwide were experiencing foodborne illnesses caused by food spoilage microorganisms. Although

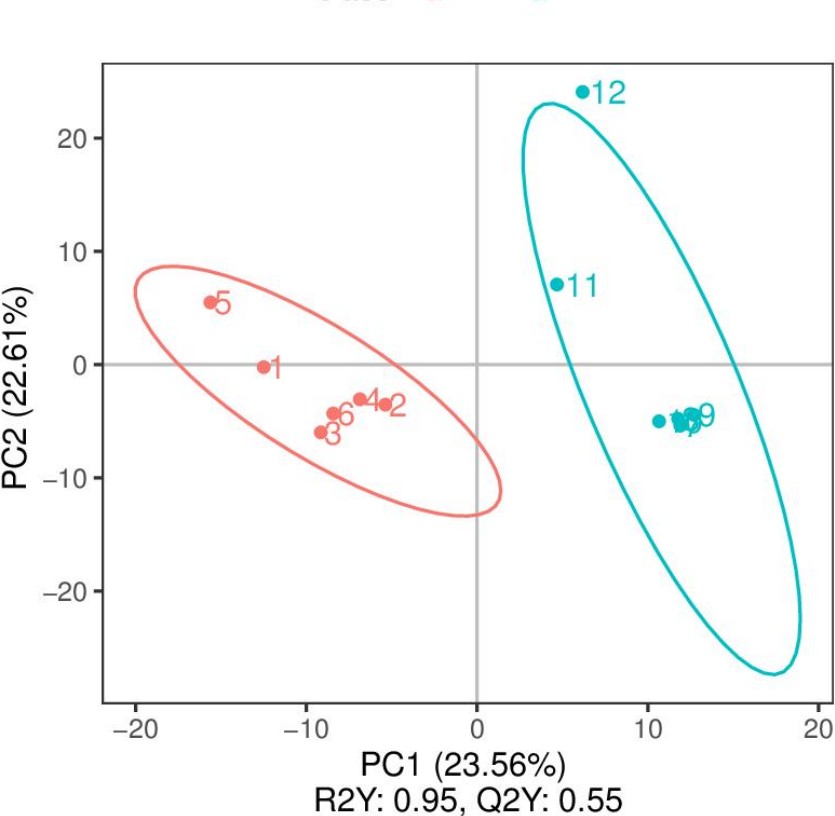

**Figure 4 Principal component analysis (PCA) analysis of D and DC samples.** The ordinate represented the clustering of samples and the abscissa represented the clustering of metabolites. D were represented Aspergillus flavus treated with sec-butylamine (1, 2, 3, 4, 5, and 6 represented sample replicates), DC represented control samples (7, 8, 9, 10, 11, and 12 represented sample replicates).

there was a wave of natural food preservation agent development at home and abroad, chemical food preservation agents remain the most widely used method worldwide, primarily due to cost and application scale limitations (*Tiwari et al., 2009*). For instance, some commonly used preservatives in the food industry, such as sodium benzoate, citral, and sodium diacetate, have a significant inhibitory effect on *A. flavus* (*Li et al., 2022a*). Other preservatives such as clove essential oil, fungil urea, and $SO_2$ fumigant had a notable inhibitory effect on *Streptomyces* spp. (*Falleh et al., 2020*). However, preservatives that had a significant inhibitory effect on *T. funiculosus* were rarely involved in the current market. Therefore, further research was needed on the development of preservatives with a significant inhibitory effect on *T. funiculosus*. It was worth noting that chemical preservatives were facing controversies due to problems of environmental pollution, and drug residues (*Haenni et al., 2022*). Therefore, precise control of the usage, dosage, and residual amount of chemical preservatives in practical applications was crucial in addressing these issues. The author conducted this experiment to provide data reference for the dosage

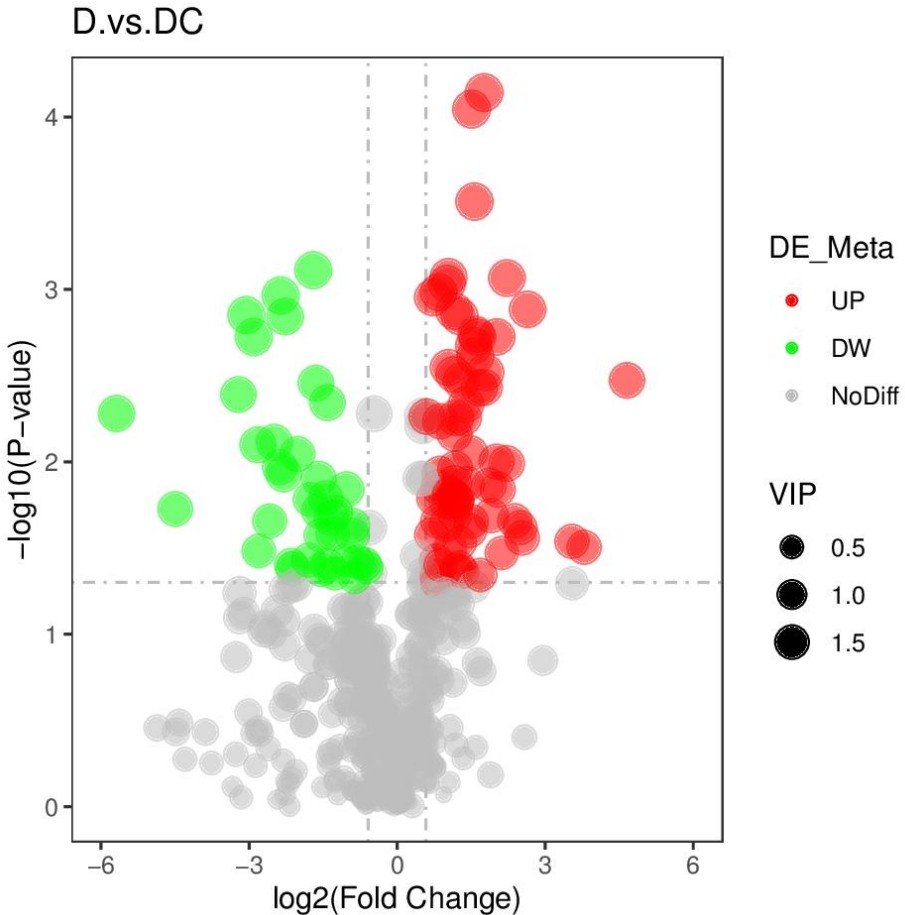

**Figure 5** **Volcano plots of differential metabolites of D and DC samples.** D represented *Aspergillus flavus* treated with sec-butylamine, and DC represented control samples.

of chemical preservatives in actual preservation and to provide data support for related research in the future, based on the above problems.

The three preservation agents used in this experiment were widely used in the food industry. Sec-butylamine was commonly used in the preservation of fruits and vegetables after harvesting and during storage (*Li et al., 2018*). Citric acid was widely used in canned food, jams, jellies, and other products. It not only inhibited the growth and reproduction of spoilage fungi but also adjusted the pH value, improving the quality and flavor of food (*Win et al., 2021*). Potassium sorbate was widely used in the food industry and the preservation of health products, animal feed, pharmaceuticals, cosmetics, paper pulp, fabrics, and more. It was less toxic and thermally stable and could maintain the nutrients and flavor of food during preservation. In recent years, it had been widely used at home and abroad (*Yardimci et al., 2022*).

However, there were some shortcomings in the practical application of these three preservation agents. Sec-butylamine could be harmful to humans when inhaled, orally
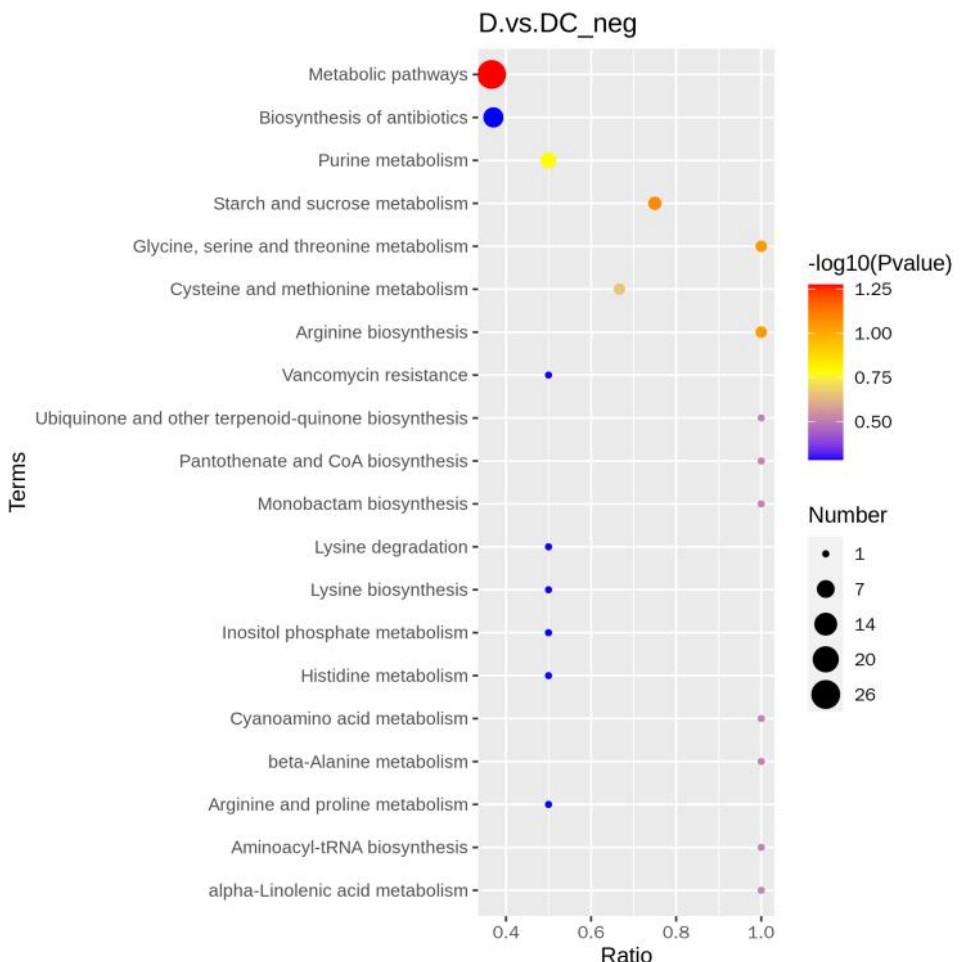

**Figure 6  KEGG enrichment map of differential metabolites.** D represented *Aspergillus flavus* treated with sec-butylamine and DC represented control samples. The horizontal coordinate indicates the rich factor of each pathway, the vertical coordinate is the pathway name, and the dot's color is the *p* value; the redder it is, the more significant the enrichment. The size of the dots represents the number of differential metabolites enriched.

ingested, or absorbed through the skin. Prolonged contact could cause serious local irritation or burns, and sec-butylamine was flammable. Its vapor was heavier than air, could diffuse at a lower diffusion, and could catch fire back to combustion when exposed to a source of fire. Care should be taken to avoid human contact (*Durgapal et al., 1987*). Citric acid affected the metabolism of calcium, and excessive intake could lead to calcium deficiency. Additionally, it could not be added to pure milk as it could cause pure milk coagulation (*Pooresmaeil et al., 2022*). Potassium sorbate was recognized as a low-toxicity, safe, and efficient food preservative. However, its preservative effect was small under neutral conditions, and it could only give full play to its preservative effect in an acidic medium (*Hameed, Xie & Ying, 2018*). Therefore, preservatives should be used in production and

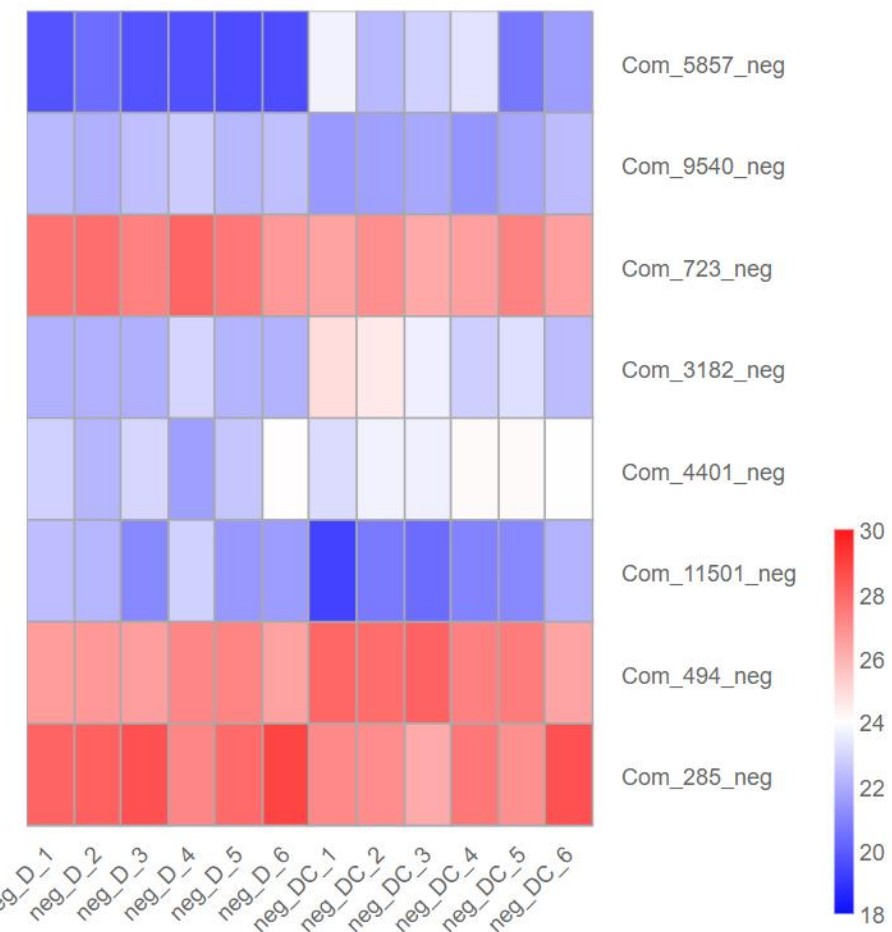

**Figure 7   Expression heatmap of differentially metabolites in sec-butylamine inhibits *Aspergillus flavus*.** D represented *Aspergillus flavus* treated with sec-butylamine, and DC represented control samples. The differential expression of metabolites in the heatmap was treated with log2.

use in a safe and regulated manner to ensure their effectiveness and reduce any potential health risks.

The research showed that the mycelium and spores were seriously distorted after the antifungal treatment (*Yahyazadeh, Omidbaigi & Taheri, 2008*). Therefore, it was speculated that sec-butylamine may destroy the structure of cell membrane, affect the normal physiological metabolism of *A. flavu* s cells, caused the outflow of cell contents, and ultimately caused the collapse and necrosis of *A. flavus*. Under the treatment of 0 and 1/2 MIC paeonol, the gene expression profile of *A. flavus* mycelia was analyzed by RNA sequencing. DEGs were involved in cell wall polysaccharide (β-1, 3-glucan, chitin) and its related enzymes synthesis, cell membrane composition (such as ergosterol, trehalose, *etc.*), REDOX system and aflatoxin biosynthesis. It mainly affected the metabolic process and cell components such as cell membrane and organelles of *A. flavus* (*Li et al., 2022b*). The expression of ergosterol (CYP51B) and four chitinases (CHSB, CHSC, CHSE,

CHSG) in *A. flavus* treated with perilla aldehyde were down-regulated. In this study, the combined analysis of DEGs and differentially expressed metabolites related to antibacterial properties showed a strong correlation. The differential expression of the four chitinases proved that perillaldehyde treatment damaged the integrity of the cell wall of *A. flavus* (*Duan et al., 2023*). The integrity of the plasma membrane was damaged, resulting in the destruction of membrane permeability, the leakage of intracellular substances, affecting the normal synthesis and metabolism of various enzymes, and blocking the metabolism of *A. flavus* (*Sant et al., 2016*). Many antibacterial products on the market, such as imidazole and triazole, were the main antifungal drugs (*Chaudhary, Tupe & Deshpande, 2013*). Destruction of cell membranes was a common strategy used as a potential fungicide, such as inhibiting the biosynthesis of membrane components, destroying membrane structural components, *etc.* (*Pan et al., 2019*). Therefore, we studied the mechanism of inhibiting *A. flavus* with sec-butylamine.

## CONCLUSIONS

The experimental results of the inhibition effect of the three preservatives on the three pathogenic fungi of *A. flavus*, *A. alternata*, and *T. funiculosus*, respectively, showed that the inhibition effect of the three preservatives on the pathogenic fungi was significant, and all of them could play an absolute inhibition effect on the pathogenic fungi of *A. flavus*, *A. alternata*, and *T. funiculosus* under a certain concentration. Under the experimental conditions, 0.6 mg/ml of sec-butylamine, 1.2 mg/ml of citric acid, and 0.2 mg/ml of potassium sorbate showed an absolute inhibitory effect on the growth of *A. flavus*. 0.5 mg/ml of sec-butylamine, 1.0 mg/ml of citric acid, and 0.6 mg/ml of potassium sorbate showed an absolute inhibitory effect on the growth of *A. alternata*. 1.0 mg/ml of sec-butylamine, 1.2 mg/ml of citric acid, and 0.8 mg/ml of potassium sorbate showed an absolute inhibition on the growth of *T. funiculosus*. However, the bioassay results of these three preservatives, the residual amount of preservatives, and other issues have not been studied clearly. Currently, only a single indoor microbial inhibition effect study of these three preservatives has been conducted. The key genes of sec-butylamine inhibits *A. flavus*, the expression of AFLA_053390, AFLA_121370, AFLA_024930, and AFLA_041970 were down-regulated in *A. flavus*. AFLA_002830 and AFLA_030450 were treated with sec-butylamine and their expression levels decreased in *A. flavus*. We further analyzed metabolites associated with sec-butylamine inhibits *A. flavus*. (3R)-4,4-Dimethyl-2-oxotetrahydro-3-furanyl beta-D-glucopyranoside (Com_5857_neg), Trehalose (Com_3182_neg) , D-Glucosamine 6-phosphate (Com_4401_neg), as well as Sucrose (Com_494_neg) present at high levels after sec-butylamine treatment in *A. flavus*. In future research, other preservatives or their mixed use could be considered to enhance the inhibition effect. These issues needed further exploration and research.

## ACKNOWLEDGEMENTS

The authors are grateful to the Hebei Provincial Technology Innovation Center for High-value Utilization of Edible and Medicinal Fungi Resources.

## Funding

The present study was funded by the special Project of Hebei Province Key Research and Development Plan Project on Rural Revitalization Technology Innovation (22327103D), the PhD (post) initiation fund of Langfang Normal University (XBQ202309), Scientific research project of colleges and universities in Hebei Province (QN2025091), and Self-finance Project of Science, and Technology Bureau of Langfang City, Hebei Province, China (2024013120). The funders had no role in study design, data collection and analysis, decision to publish, or preparation of the manuscript.

## Grant Disclosures

The following grant information was disclosed by the authors:
Hebei Province Key Research and Development Plan Project on Rural Revitalization Technology Innovation: 22327103D.
Langfang Normal University: XBQ202309.
Scientific research project of colleges and universities in Hebei Province: QN2025091.
Self-finance Project of Science, and Technology Bureau of Langfang City, Hebei Province, China: 2024013120.

## Competing Interests

The authors declare there are no competing interests.

## Author Contributions

- Zhenxia Shi conceived and designed the experiments, performed the experiments, prepared figures and/or tables, and approved the final draft.
- Ni Zhan conceived and designed the experiments, performed the experiments, prepared figures and/or tables, authored or reviewed drafts of the article, and approved the final draft.
- Ming Ma performed the experiments, prepared figures and/or tables, authored or reviewed drafts of the article, and approved the final draft.
- Zhen Wang conceived and designed the experiments, performed the experiments, prepared figures and/or tables, and approved the final draft.
- Xunyou Yan conceived and designed the experiments, authored or reviewed drafts of the article, and approved the final draft.
- Rumeng Li analyzed the data, prepared figures and/or tables, and approved the final draft.
- Xuejuan Liu analyzed the data, prepared figures and/or tables, and approved the final draft.

## DNA Deposition

The following information was supplied regarding the deposition of DNA sequences:
The RNA-Seq reads are available at Sequence Read Archive (SRA): PRJNA1169418.
The metabolome data are available at National Genomics Data Center (NGDC): PRJCA032029 (OMIX007803).

## Data Availability

The RNA-Seq reads are available at Sequence Read Archive (SRA): PRJNA1169418.

## Supplemental Information

Supplemental information for this article can be found online at http://dx.doi.org/10.7717/peerj.19737#supplemental-information.

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
