# Peer review of "Transcriptome and metabolome profiling reveal the inhibitory effects of food preservatives on pathogenic fungi"

_PeerJ, doi:10.7717/peerj.19737_

## Round 0.1 · original submission · Major Revisions

Shi et al.'s manuscript explores the inhibitory effects of preservatives on foodborne fungi through transcriptomic and metabolomic analyses. Although it includes considerable experimental work, the manuscript needs improvement due to notable problems in its structure, writing quality, and scientific rigor. In light of these challenges, we request major corrections; the study necessitates significant restructuring, reanalysis, and rewriting.

·

Basic reporting

The study by Shi et al, investigated the inhibitory roles of sec-butylamine, potassium sorbate, and citric acid on three food spoilage fungi. The authors further use RNA sequencing and metabolite analysis to know the inhibitory mechanism of sec-butylamine against. The findings are scientifically interesting and potentially valuable to the field of food preservation. However, the manuscript suffers from structural issues, unclear language in places, and insufficient depth in data interpretation.

Experimental design

The manuscript examines three preservatives and three fungi, but the focus appears scattered. To enhance clarity and impact, I recommend narrowing the scope to focus solely on A. flavus and its interactions with sec-butylamine. This will streamline the study and make the findings more impactful. Data on A. alternata and T. funiculosus could be relegated to supplementary material.

Validity of the findings

The findings presented in this manuscript are supported by a combination of experimental approaches, including antifungal activity assays, transcriptome analysis, and metabolome profiling. The experimental design is generally sound, and the methodologies used are appropriate for studying the inhibitory effects of preservatives on foodborne fungi. However, the interpretation of the results lacks depth, particularly in linking the transcriptomic and metabolomic data to the antifungal mechanisms.

Additional comments

Some suggestions are indicated as below:
1. The authors have used tables to present results; however, bar graphs or other visual figures would be more effective in communicating trends and differences in the data.
2. Figures 3–5 should be combined into one composite figure for better visualization and easier comparison.
3. The individual effects of these preservatives on foodborne fungi have been reported previously. The authors can address this by investigating the combined inhibitory effects of the preservatives, which would offer new insights.
4. The discussion of transcriptomic and metabolomic data lacks depth. A more comprehensive understanding of the preservative's inhibitory mechanism should be addressed. The authors can also analyze the metabolic pathways involved in aflatoxin production in A. flavus and how sec-butylamine impacts these pathways.
5. In the introduction section, the safe usage limits of the three preservatives should be clearly stated to provide context for their application in food systems.
6. Scientific names of microorganisms (e.g., Aspergillus flavus, Alternaria alternata, Talaromyces funiculosus) must be italicized throughout the manuscript.
7. Upon first mention, both genus and species names should be written in full. Subsequently, abbreviate the genus (e.g., A. flavus). Ensure this is consistently applied.
8. Lines 29–32: The sentences in this section are unclear and need revision for clarity.
9. Lines 64–66: Correct spacing is required around some commas.
10. Line 88: Include the sources and strain numbers for the microorganisms used.
11. Line 90: The terms “suspensions and cakes” are ambiguous; please clarify.
12. Line 94: “6 mm”.
13. Line 98: Revise to "10^5–10^6 spores/mL".
14. Lines 102–103: Format volumes as “2 mL,” “100 mL,” etc., consistently.
15. Line 200: The organisms discussed are fungi, not bacteria. Correct the terminology.
16. Line 218: Remove the text “Add your discussion here.”

Reviewer 2 ·

Basic reporting

It is a very interesting study. You could improve the quality of your job if you improve the grammar and the style of the English you use. In some paragraphs, you use active voice; in others, you use passive voice. I strongly suggest a spelling check for the complete document and a correction style to homogenize the language used in the paper.
You refer to studied microorganisms as “bacteria” in several parts of the document; that’s not correct; please avoid this kind of expression to make it clearer for the reader.

Experimental design

The study defines the aim of the research; maybe if you define the acronym used the first time you mentioned it, especially in the molecular section, it will be easier to follow the text. Some important details should be added to ensure someone else can replicate your experiments, for instance, usually, you add tartaric acid in PDA to avoid bacterial contamination; it is not clear if you only added your preservatives tested or also you added tartaric acid. You should also specify if different incubations were at the same temperature or under the same conditions. Some specific observations are included in the additional comments session of this revision.

Validity of the findings

no comment

Additional comments

Line 48 and 49, you mention that Alternaria alternata and Aspergillus flavus are bacteria; both microorganisms are molds.
All scientific names of microorganisms should be written in cursive letters.
Line 55 Talaromyces funiculosus is also a fungi, not a bacteria.
Line 56, the word “strains” could be added after “air and soil, some …”; “of which” could be removed, and also you may change “human” to “industrial”
Line 57, “in industrial production” could be removed.
Line 58, you need to add a space between “;” and some. Also you should change the bacteria word. Also, “and deterioration” could be removed because you include that effect using the word spoilage.
Line 82, please replace “bacteria” by “fungi” or “molds”.
Line 94, please add a space between 6 and mm.
Line 95, shaken conditions should be explained, not only shaken time.
Line 98, Is it correct 105-106 CFU/mL or is a scientific notation of 100,000 and 1,000,000 spores/mL
Line 100, you may change the word “for” to “during.”
Line 102, and space should be added between 2 and ml.
Line 103, and space should be added between 100 and ml.
Line 102-107, you should consider the following text to describe the experiment: Inhibition effect of preservatives on the test strains: 2 ml of different concentrations of food preservative solution were mixed with 100 ml of PDA medium and poured into sterile Petri dishes, the medium was retained for naturally cooling and put in the incubator, 28 °C, during 24 hours to ensure sterility of plate. Once incubation time was reached and no growth of the miscellaneous fungus was observed, the suspension of the tested strain was inoculated on the Petri plate and incubated at 28 °C for 36 hours in order to observe fungus growth.
Line 109, a space should be added between 100 and ml. The use of passive voice should be considered for the experiment.
Line 112, please replace “bacterium cake” by “fungi/mold cake”.
Lines 102-116, you should clarify the difference between both experiments; they look similar.
Line 117 and 120 Is it the same equation? Please clarify the difference between them or eliminate one of them.
Line 122, virulence could be replaced by inhibition.
Line 123, virulence could be replaced by inhibition.
Line 124, bacteria should be replaced by fungi.
Line 128, bacteria should be replaced by fungi.
Line 131, please replace "for" with "during" when it refers to time.
Line 136, maybe you should change “sprayed” to “sprayed the samples.”
Line 155, a space should be added after “was”.
Line 159, a space should be added after “qRT-PCR:”.
Line 200, avoid the use of the word “bacteria”, instead you may use “fungi”.
Line 207-217, it is not clear if pathogenic bacteria refer to the microorganisms in this study oor if you are talking about other microorganisms. It would be best if you clarified your comments.
Line 218, Why is this note here?
Line 241 and 244, the word “bacteria” needs to be replaced.
Line 255, 256 and 258, “aflatus” means “flavus”? it would be best if you changed the species name.
Line 267, please you need to correct the word “flavusn”.
Line 271, high quality of the sequencing data… according to? (if is possible, you should provide a reference)
Line 279 and 285, the meaning of DEGs and KEGG should be presented at least the first time they were mentioned.
Line 310, please remove the first “and”
Line 322, please, you could not assume that bacteria is synonymous with microorganisms.
Line 327, microbial contamination of fresh vegetables is a worldwide problem, not just for China, you could add some other references and change this paragraph to increase the impact of your findings.
Line 333, please add “spp.” Or some specific name after “Streptomyces”.
Line 338, pathogenic bacteria resistance is a different problem; maybe it is unrelated to your study.
In line 355, you should specify normal temperatures' meaning or range values.
Line 359, could you be more specific about “can affect health”?
Line 366, which antibacterial treatments are you talking about?
Line 369, please send the capital letter for Aspergillus.
Line 388-390, which pathogenic bacteria are you talking about?
Page 22, line 1, please replace “secbutamide” for “sec-butylamine”
Page 22, line 3, you should avoid using “bacterium”.
Page 24, line 3, you should avoid using “bacterium”.
Page 26, line 3, you should avoid using “bacterium”.
Table 4-6, you should avoid the use of “bacteria”
Table 7-9, you should replace “secbutamide” and refer to the equation as “Toxicity” instead of “virulence”. Also you should define the meaning of “y” and “x” in the same equation.

---

## Round 0.2 · Major Revisions

Please consider the reviewers' suggestions and submit a revised version along with a point-by-point response letter that addresses all concerns.

·

Basic reporting

The authors have made significant efforts to revise the manuscript. However, there are still considerable issues in the structure and presentation of the results. Tables 1-3 do not effectively convey their contents, and I recommend that the authors present the data in bar charts for better clarity. Figures 2-5, which relate to transcriptomic results, do not provide substantial insights. I suggest that the authors combine these figures into a single panel or move them to the supplementary materials. Additionally, the transcriptomic and metabolomic analyses lack in-depth interpretation and discussion. I recommend that the authors provide a more thorough analysis of these results to enhance the overall clarity and impact of the study.

Experimental design

The experimental design appears to be well thought out, as the authors have selected three common preservatives (sec-butylamine, potassium sorbate, and citric acid) and tested their inhibitory effects on multiple plant pathogens, including Aspergillus flavus, Alternaria alternata, and Talaromyces funiculosus. The use of both transcriptomic and metabolomic analyses to explore the inhibitory mechanism of sec-butylamine on A. flavus is a good approach for identifying key molecular changes and gaining a deeper understanding of its effects. However, the study would benefit from clearer presentation and more detailed analysis of the results, particularly in terms of the biological relevance of the identified genes and metabolites.

Validity of the findings

This study appears to be generally sound, as the study provides comprehensive data on the inhibitory effects of sec-butylamine, potassium sorbate, and citric acid on various fungal pathogens. The experimental conditions, such as the concentrations of the preservatives, seem appropriate for testing their effectiveness against Aspergillus flavus, Alternaria alternata, and Talaromyces funiculosus. Furthermore, the use of transcriptomic and metabolomic analyses to explore the mechanism of action for sec-butylamine is a valuable approach. However, the lack of in-depth analysis and interpretation of the transcriptomic and metabolomic results limits the overall strength of the findings. Additionally, the presentation of results could be clearer and more concise, which would enhance the reliability and impact of the conclusions.

Additional comments

The suggestions can be found in the Basic reporting.

Reviewer 2 ·

Basic reporting

The English used has improved substantially compared to the previous version of the article. A minor second edit is recommended to improve the quality of the article further and prepare it for publication. The reference list is appropriate and in line with the topic presented. The results are consistent with the implicit hypothesis.

Experimental design

The work is aligned with the journal's objectives and scope. The research question, although implicit, is straightforward. The methodology described is sufficient.

Validity of the findings

The data presented as results are clear and have a good level of analysis and statistical significance. The key raw data were provided, and good conclusions can be drawn from these results.

Additional comments

The comments were promptly addressed, and only minor changes were required for the English language edition.

---

## Round 0.3 · accepted · Accept

All issues pointed out by the reviewers were addressed, and the revised manuscript is acceptable now.